# Constraints on the spectrum of field theories with non-integer $O(N)$ symmetry from quantum evanescence

Weiguang Cao,[1, 2] Xiaochuan Lu,[3] and Tom Melia[1]

[1]*Kavli Institute for the Physics and Mathematics of the Universe (WPI),*
*UTIAS, The University of Tokyo, Kashiwa, Chiba 277-8583, Japan*
[2]*Department of Physics, Graduate School of Science,*
*The University of Tokyo, Tokyo 113-0033, Japan*
[3]*Department of Physics, University of California, San Diego, La Jolla, CA 92093, USA*

We identify constraints in the energy spectra of quantum theories that have a global $O(N)$ symmetry, where $N$ is treated as a continuous parameter. We point out that a class of evanescent states fall out of the spectrum at integer values of $N$ in pairs, via an annihilation mechanism. This forces the energies of the states in such a pair to approach equality as $N$ approaches a certain integer, with both states disappearing at precisely integer $N$ and the point of would-be degeneracy. These constraints occur between different irreducible representations of the analytic continuation of $O(N)$ and hold non-perturbatively. We give examples in the spectra of the critical $O(N)$ model.

## I. INTRODUCTION

The early models of the Hydrogen atom that ushered in the era of quantum physics exhibited a particularly striking feature: degeneracies in the energy spectrum between states of different angular momentum [1–3]. Since its subsequent symmetry-based explanation [4], the understanding of degeneracies and how they may be lifted has played a central role in the development of quantum theory. The establishment of quantum electrodynamics was driven by its exquisitely precise prediction of the Lamb shift between the nearly degenerate $2S$ and $2P$ states of Hydrogen. The large degeneracy of Landau levels plays a key role in determining the physics of the quantum Hall effect [5, 6] that marked the birth of the field of topological quantum materials [7]. Spin ice, and other geometrically frustrated systems, derive many of their properties such as emergent photon modes from the high degeneracy of their ground state [8].

Energy spectra depend on the couplings of the theory–external parameters which, if tuned, may lift or impose degeneracies, in many cases triggering a major physical change in a system. For instance, in chemical crystals known as single molecule magnets, tuning a magnetic field through points of degeneracy in a double-well energy spectrum enhances quantum tunneling between states and leads to macroscopic quantum effects in the hysteresis curve [9]. Because of the discrete energy spectrum, this phenomenon occurs at isolated values of the continuous coupling parameter.

Recently, the understanding of what qualifies as a symmetry in quantum theories has been undergoing a radical development [10]. The concept of a global symmetry $O(N)$, where $N$ is taken to be an integer, can be generalized to continuous, non-integer $N$ as a categorical symmetry [11, 12], where $N$ should be considered as a parameter of the theory. Conformal field theories with categorical $O(N)$ symmetry are known to be non-unitary for non-integer $N$ [12], and concrete physical examples where $N$ appears explicitly as a continuous coupling constant

are given by statistical $O(N)$ loop models [12–14].

In this paper, we identify constraints on the energy spectrum $\{\Delta(N)\}$ of theories with a global $O(N)$ symmetry, where $N$ is treated as a continuous parameter of the theory, and assuming spectrum continuity. These constraints set equal the energies assigned to certain pairs of states when $N$ approaches a positive integer, and in an echo of the degeneracies of the Hydrogen model, the states are in different irreducible representations (irreps) of the continuation of the $O(N)$ representation theory.

A key result of this paper is the identification of the fact that certain (but not all) evanescent states of a $O(N)$ global symmetry actually drop out of the spectrum in pairs. We show that the continued representation theory of $O(N)$ dictates that one of the evanescent states contributes to the partition function negatively, and must therefore be canceled by a positive contribution from another evanescent state. This required annihilation gives rise to a constraint on their energies being equal. That is, we argue that such states become arbitrarily close in energy as $N$ approaches an integer. At the point of would-be degeneracy - precisely integer $N$ - both states disappear from the spectrum.

Just as 'usual' degeneracy drives the physical properties and initiates macroscopic changes in many systems, one might expect this new phenomena of 'evanescent-degeneracy' (i.e. states becoming degenerate and simultaneously dropping out of the spectrum) to manifest itself in terms of physical properties. And indeed, in known cases, these points of evanescent-degeneracy (or, more simply, integer $N$) are accompanied by a major change in the physical properties of the theory: it becomes unitary. For example, the existence of unitary islands of the critical $O(N)$ model in $d = 3$ dimensions with $N = 1, 2, 3$ has been confirmed in a bootstrap approach [15, 16]. It is also known that the non-unitary nature of the theory at non-integer dimensions is tied to evanescent states [17], which become null at integer $d$.

We illustrate our findings in the critical $O(N)$ model in $4 - \epsilon$ dimensions, defined via the renormalization group

flow induced by a $\phi^4$ perturbation away from the free theory of $O(N)$-fundamental scalar fields [18]. The examples exhibit the constraints on the spectrum as a function of $N$ (we postpone a study of constraints associated with evanescent operators of space-time symmetry [17]). We work at leading perturbative order in $\epsilon$, where a wealth of data is available, e.g. [19], but we emphasize that the constraints on the spectrum as a function of $N$ in general hold at the non-perturbative level, and conformal symmetry is not a requisite.

Finally, even if we are only interested in the physics of integer $N$, it is still often useful to consider $N$ as a parameter e.g. in perturbation theory, or in $1/N$ expansions. Perhaps the most notable example of this is when we treat the dimension of space-time symmetry as a parameter: dimensional regularization [20, 21] forms the backbone of the most precise theoretical calculations ever performed. Our results have bearing on these computational frameworks, too.

We proceed as follows. In Sec. II we explain the analytic continuation of the $O(N)$ representation theory at the level of character theory and the partition function. In Sec. III we detail the effects of evanescence has in constraining the physical spectrum, and give examples in the $O(N)$ model. We conclude in Sec. IV with an outlook.

## II. ANALYTIC CONTINUATION

We proceed to write down the analytically continued partition function for a theory with an $O(N)$ global symmetry. The partition function includes fugacities to keep track of all possible $U(1)$ "good quantum numbers" of the $O(N)$ symmetry (i.e. the $m$ quantum numbers for Hydrogen), in exactly the same way as those introduced in the grand canonical partition function for $U(1)$ particle number. These fugacities then allow for keeping track of the contributions of states in different $O(N)$ irreps via a highest weight procedure: those of states belonging to the same $O(N)$ irrep factor into a function known as the Weyl character of $O(N)$. In the below, we build upon the studies of continuing the $O(N)$ representation theory in [11, 12], and focus on continuing the characters, following [22], and the partition function. We first discuss the $SU(N)$ case to illustrate the idea, and then come back to the more complicated $O(N)$ case.

**$SU(N)$ character continuation:** For linearly reductive groups, tensor products of irreps form representations, which can be decomposed back into irreps:

$$R_1 \otimes R_2 = \bigoplus_R m_{R_1 R_2}^R \, R \, . \quad (1)$$

Here $R_1, R_2$ are two arbitrary irreps, the sum runs over all irreps $R$ of the group, and $m_{R_1 R_2}^R$ is the multiplicity.

In the case of $SU(N)$, all unitary irreps can be labeled by the partitions $\lambda = (\lambda_1, \cdots, \lambda_l)$, with the length not exceeding the rank of the group: $l(\lambda) = l \leq r = N - 1$. We take the convention of not writing zeros at the end

| $\overline{\chi}_\lambda^{SU(N)}$ | $N=2$ | $N=3$ | $N=4$ | $N=5$ |
|---|---|---|---|---|
| $(1,1)$ | $\chi_{()}^{SU(N)}$ | $\chi_{(1,1)}^{SU(N)} \to$ | | |
| $(2,1)$ | $\chi_{(1)}^{SU(N)}$ | $\chi_{(2,1)}^{SU(N)} \to$ | | |
| $(3,1)$ | $\chi_{(2)}^{SU(N)}$ | $\chi_{(3,1)}^{SU(N)} \to$ | | |
| $(2,2)$ | $\chi_{()}^{SU(N)}$ | $\chi_{(2,2)}^{SU(N)} \to$ | | |
| $(1,1,1)$ | $0$ | $\chi_{()}^{SU(N)}$ | $\chi_{(1,1,1)}^{SU(N)} \to$ | |
| $(2,1,1)$ | $0$ | $\chi_{(1)}^{SU(N)}$ | $\chi_{(2,1,1)}^{SU(N)} \to$ | |
| $(1,1,1,1)$ | $0$ | $0$ | $\chi_{()}^{SU(N)}$ | $\chi_{(1,1,1,1)}^{SU(N)} \to$ |

TABLE I. Example specializations of the continued character $\overline{\chi}_\lambda^{SU(N)}$ in terms of the ordinary character $\chi_{\lambda'}^{SU(N)}$. The arrow '$\to$' denotes that the entry is repeated to complete the row.

| $\overline{\chi}_\lambda^{O(N)}$ | $N=2$ | $N=3$ | $N=4$ | $N=5$ | $N=6$ |
|---|---|---|---|---|---|
| $(1,1)$ | $\chi_{()}^{O(N)}$ | $\chi_{(1)}^{O(N)}$ | $\chi_{(1,1)}^{O(N)} \to$ | | |
| $(2,2)$ | $-\chi_{(2)}^{O(N)}$ | $0$ | $\chi_{(2,2)}^{O(N)} \to$ | | |
| $(2,1,1)$ | $-\chi_{()}^{O(N)}$ | $0$ | $\chi_{(2)}^{O(N)}$ | $\chi_{(2,1)}^{O(N)}$ | $\chi_{(2,1,1)}^{O(N)} \to$ |
| $(2,2,1)$ | $-\chi_{(1)}^{O(N)}$ | $-\chi_{(2)}^{O(N)}$ | $0$ | $\chi_{(2,2)}^{O(N)}$ | $\chi_{(2,2,1)}^{O(N)} \to$ |
| $(2,2,2)$ | $-\chi_{()}^{O(N)}$ | $-\chi_{(2)}^{O(N)}$ | $-\chi_{(2,2)}^{O(N)}$ | $0$ | $\chi_{(2,2,2)}^{O(N)} \to$ |
| $(3,2,1)$ | $0$ | $-\chi_{(3)}^{O(N)}$ | $0$ | $\chi_{(3,2)}^{O(N)}$ | $\chi_{(3,2,1)}^{O(N)} \to$ |
| $(3,3,1)$ | $\chi_{(3)}^{O(N)}$ | $0$ | $0$ | $\chi_{(3,3)}^{O(N)}$ | $\chi_{(3,3,1)}^{O(N)} \to$ |

TABLE II. Example specializations of the continued character $\overline{\chi}_\lambda^{O(N)}$ in terms of the ordinary character $\chi_{\lambda'}^{O(N)}$. The arrow '$\to$' denotes that the entry is repeated to complete the row.

of a partition, and the singlet irrep is therefore denoted by the length zero partition $\lambda = ()$. Each partition $\lambda$ corresponds to a Young diagram, with $\lambda_i$ boxes in the $i$-th row. The characters $\chi_\lambda^{SU(N)}(x_1, \cdots, x_r)$ are traces over the matrices of group elements, with $x_i = e^{i\theta_i}$ the fugacities parameterizing the $U(1)^r$ Cartan subgroup of $SU(N)$ (i.e. the "good quantum numbers"); see e.g. [23]. In terms of the characters, the tensor product decomposition algebra in Eq. (1) reads

$$\chi_{\lambda_1}^{SU(N)} \chi_{\lambda_2}^{SU(N)} = \sum_{\lambda,\, l(\lambda) \leq r} m_{\lambda_1 \lambda_2}^\lambda(N) \, \chi_\lambda^{SU(N)} \, . \quad (2)$$

For any given irreps $\lambda_1, \lambda_2$, only a finite number of irreps $\lambda$ in this sum will have nonzero multiplicities, as the product corresponds to a finite dimensional representation. The multiplicities $m_{\lambda_1 \lambda_2}^\lambda(N)$ vary with $N$, but have asymptotic values for sufficiently large finite $N$, known as the Newell-Littlewood numbers (see e.g. [24]):

$$\overline{m}_{\lambda_1 \lambda_2}^\lambda \equiv \lim_{N \to \infty} m_{\lambda_1 \lambda_2}^\lambda(N) \, . \quad (3)$$

Therefore, the large $N$ limit of the algebra in Eq. (2)

holds for all $N$ in the same way, providing a natural continuation of it to non-integer $N$ [11, 12, 21]:

$$\overline{\chi}_{\lambda_1}^{SU(N)} \, \overline{\chi}_{\lambda_2}^{SU(N)} = \sum_{all \ \lambda} \overline{m}_{\lambda_1 \lambda_2}^{\lambda} \, \overline{\chi}_{\lambda}^{SU(N)} \,, \qquad (4)$$

where we introduced $\overline{\chi}_{\lambda}^{SU(N)}$ which we call the continued character, while referring to $\chi_{\lambda}^{SU(N)}$ as the ordinary character. Some simple examples of the continued algebra in Eq. (4) are

$$\square \otimes \square = \square\square \oplus \begin{array}{c}\square\\\square\end{array} \,, \qquad (5a)$$

$$\square \otimes \square \otimes \square = \square\square\square \oplus 2\begin{array}{c}\square\square\\\square\end{array} \oplus \begin{array}{c}\square\\\square\\\square\end{array} \,, \qquad (5b)$$

$$\square \otimes \square \otimes \square \otimes \square = \square\square\square\square \oplus 3\begin{array}{c}\square\square\square\\\square\end{array} \oplus 2\begin{array}{c}\square\square\\\square\square\end{array} \oplus 3\begin{array}{c}\square\square\\\square\\\square\end{array} \oplus \begin{array}{c}\square\\\square\\\square\\\square\end{array} \,. \qquad (5c)$$

At a given finite integer $N$, the continued characters $\overline{\chi}_{\lambda}^{SU(N)}$ will "specialize" [22] as zero or as some ordinary ones, with the effect that the continuation in Eq. (4) specializes as Eq. (2). The specialization rules are

$$\overline{\chi}_{\lambda=(\lambda_1,\cdots,\lambda_l)}^{SU(N)} = \begin{cases} \chi_{\lambda}^{SU(N)} & N > l \\ \chi_{(\lambda_1-\lambda_l,\cdots,\lambda_{l-1}-\lambda_l)}^{SU(N)} & N = l \\ 0 & N < l \end{cases} \,, \qquad (6)$$

with some explicit examples listed in Table I. The rule for $N = l$ is recognizable as originating from the contraction of the irrep with the $SU(N = l)$ epsilon tensor. Using these, one can check that at $N = 2$, Eq. (5) does specialize to reproduce the expected ones for $SU(2)$:

$$\mathbf{2} \times \mathbf{2} = \mathbf{3} + \mathbf{1} \,, \qquad (7a)$$

$$\mathbf{2} \times \mathbf{2} \times \mathbf{2} = \mathbf{4} + 2\,(\mathbf{2}) \,, \qquad (7b)$$

$$\mathbf{2} \times \mathbf{2} \times \mathbf{2} \times \mathbf{2} = \mathbf{5} + 3\,(\mathbf{3}) + 2\,(\mathbf{1}) \,, \qquad (7c)$$

while for $SU(3)$, Eq. (5b) does specialize as

$$\mathbf{3} \times \mathbf{3} \times \mathbf{3} = \mathbf{10} + 2\,(\mathbf{8}) + \mathbf{1} \,. \qquad (8)$$

From the Young diagram point of view, Eq. (6) can be described as clipping off the leftmost columns (the "West Coast") that have $N$ boxes. If these columns have more than $N$ boxes, the clipping fails and returns zero. More details are provided in the appendix.

**$O(N)$ character continuation:** In this paper, we focus on $O(N)$ irreps with integer spins. They can also be labeled by partitions $\lambda = (\lambda_1, \cdots, \lambda_l)$ (and hence Young diagrams), again with the length $l$ not exceeding the rank $r$ of the group, which is now given by $r = \lfloor N/2 \rfloor$. For partitions with $2l < N$, we take it to mean the parity even representation of $O(N)$. The ordinary characters $\chi_{\lambda}^{O(N)}(x_1, \cdots, x_r)$ can be found in e.g. [25].

The same procedure of continuing the tensor product decomposition algebra from Eq. (2) to Eq. (4) holds for the $O(N)$ case. However, the specialization rule of the continued character $\overline{\chi}_{\lambda}^{O(N)}$ in terms of the ordinary ones $\chi_{\lambda'}^{O(N)}$ is not as simple as in Eq. (6). The new rule can be worked out by considering the vector representation, which is valid for any integer $N \geq 1$ (for $N = 1$ it is the character for the trivial irrep of the $Z_2$ symmetry):

$$\chi_{(1)}^{O(N)}(x_1, \cdots, x_r) = \frac{1 - (-1)^N}{2} + \sum_{i=1}^{r} (x_i + x_i^{-1}) \,. \qquad (9)$$

The continued characters can be computed from it as

$$\overline{\chi}_{\lambda}^{O(N)}(x) = F_{\lambda}\left[ \chi_{(1)}^{O(N)}(x), \cdots, \chi_{(1)}^{O(N)}(x^q) \right] \,, \qquad (10)$$

where $q = \lambda_1 + \cdots + \lambda_l$ and we are using the shorthand $x = (x_1, \cdots, x_r)$ and $x^k = (x_1^k, \cdots, x_r^k)$. The point is that the functions $F_{\lambda}$ are independent of $N$, encoding the Newell-Littlewood numbers $\overline{m}_{\lambda_1 \lambda_2}^{\lambda}$ in the continued tensor product decomposition algebra. Their explicit expressions are known [22], which we also reproduce in the appendix.

For example, when $\lambda = (1,1)$, the explicit form of $F_{(1,1)}$ reads

$$\overline{\chi}_{(1,1)}^{O(N)}(x) = \tfrac{1}{2}\left[ \chi_{(1)}^{O(N)}(x) \right]^2 - \tfrac{1}{2} \chi_{(1)}^{O(N)}(x^2) \,. \qquad (11)$$

For $N \geq 4$, $\lambda = (1,1)$ gives a valid representation, and hence $\overline{\chi}_{(1,1)}^{O(N\geq4)}(x) = \chi_{(1,1)}^{O(N\geq4)}(x)$. At $N = 3$, Eq. (11) leads to

$$\overline{\chi}_{(1,1)}^{O(3)}(x) = 1 + x_1 + x_1^{-1} = \chi_{(1)}^{O(3)}(x) \,, \qquad (12)$$

while at $N = 2$ it gives

$$\overline{\chi}_{(1,1)}^{O(2)}(x) = 1 = \chi_{()}^{O(2)}(x) \,. \qquad (13)$$

Similarly, the explicit form of $F_{(2,2)}$ and $F_{(2,1,1)}$ give

$$\overline{\chi}_{(2,2)}^{O(2)}(x) = -x_1^2 - x_1^{-2} = -\chi_{(2)}^{O(2)}(x) \,, \qquad (14a)$$

$$\overline{\chi}_{(2,2)}^{O(1)} = -1 = -\chi_{()}^{O(1)} \,, \qquad (14b)$$

$$\overline{\chi}_{(2,1,1)}^{O(2)}(x) = -1 = -\chi_{()}^{O(2)}(x) \,. \qquad (14c)$$

A few more nontrivial examples are summarized in Table II. Compared with Table I, we see that a prominent new feature of the $O(N)$ case is the negative signs. We will see in Sec. III that these lead to the constraints.

The above provides an explicit method for calculating how a given continued character $\overline{\chi}_{\lambda}^{O(N)}$ will specialize as an ordinary one $\chi_{\lambda'}^{O(N)}$ with $l(\lambda') \leq r$. This can also be achieved by the method of "folding a Young diagram" described in [22], or our alternative prescription of "clipping the East Coast of a Young diagram" described in the appendix.

**Partition function continuation:** With the continuation of the characters and a continued tensor product

decomposition algebra (e.g. Eq. (4)) that holds for all $N$ obtained, we are now able to write down a continuation of the partition function that is valid for all values of real $N$, integer or non-integer,

$$\overline{Z}\left(q, \{\Delta(N)\}\right) = \sum_{all\ \lambda} \sum_{i=1}^{n(\lambda)} q^{\Delta_{\lambda,i}(N)} \overline{\chi}_\lambda^{O(N)}. \qquad (15)$$

As in Eq. (4), the sum over $\lambda$ is over all irreps, namely all partitions without restriction on the length. The sum $i$ runs across all the states in a given irrep $\lambda$, with their total number $n(\lambda)$ playing the role of the multiplicities $\overline{m}_{\lambda_1\lambda_2}^\lambda$. The $\Delta_{\lambda,i}(N)$ denote energies of the states, and are functions of $N$.

## III. SPECTRUM CONSTRAINTS

With the analytic continuation of the partition function in Eq. (15), we are now able to state the nature of the constraints on the physical spectrum of theories with analytically continued global $O(N)$ symmetry.

At each integer $N$, the continued characters for irreps of length $l > r = \lfloor N/2 \rfloor$ will specialize as zero, or as some valid irrep characters with $l(\lambda) \leq r$, possibly with an overall minus sign; see Table II for explicit examples. We denote irreps in these different cases by $\lambda^0$ and $\lambda^\pm$, respectively. That is,

$$\overline{\chi}_{\lambda^0}^{O(N)} = 0, \qquad \overline{\chi}_{\lambda^\pm}^{O(N)} = \pm\chi_\lambda^{O(N)}. \qquad (16)$$

We can then write the sum in Eq. (15) at an integer $N$ with $\lambda$ running over only *valid* irreps of $O(N)$:

$$Z^{(N)}\left(q, \{\Delta(N)\}\right) = \sum_{\lambda,\,l(\lambda)\leq r} \chi_\lambda^{O(N)} \left( \sum_{i=1}^{n(\lambda)} q^{\Delta_{\lambda,i}(N)} \right.$$
$$\left. + \sum_{i=1}^{n(\lambda^+)} q^{\Delta_{\lambda^+,i}(N)} - \sum_{i=1}^{n(\lambda^-)} q^{\Delta_{\lambda^-,i}(N)} \right). \qquad (17)$$

Here $n(\lambda)$ remains the same as in Eq. (15), while $n(\lambda^\pm)$ counts the number of states with characters that specialized to $\pm\chi_\lambda^{O(N)}$.

States under the irreps $\lambda^0$ are examples of what we call direct evanescent states. They become null at the given integer $N$, contributing zero to the partition function $Z^{(N)}$, and therefore we do not derive any constraints on their spectra based on the evanescence. On the other hand, states under the irreps $\lambda^-$ give negative contributions to $Z^{(N)}$, which must be canceled by one of the original $\lambda$ states, or otherwise a $\lambda^+$ state. Therefore, at this value of $N$, we have $n(\lambda^-)$ constraints on the energy spectra:

$$\Delta_{\lambda^-,i}(N) = \Delta_{\lambda,j}(N) \quad \text{or} \quad \Delta_{\lambda^+,k}(N), \qquad (18)$$

for each $1 \leq i \leq n(\lambda^-)$.

| Lorentz irrep | $O(N)$ irrep $\lambda$ | $\Delta(N)$ |
|:---:|:---:|:---:|
| () | (2,2) | $6 - 2\epsilon + \frac{6}{N+8}\epsilon + \mathcal{O}(\epsilon^2)$ |
| () | () | $6 - 2\epsilon + \frac{2(N+2)}{N+8}\epsilon + \mathcal{O}(\epsilon^2)$ |
| (2) | (2,2) | $6 - 2\epsilon + \frac{14}{3(N+8)}\epsilon + \mathcal{O}(\epsilon^2)$ |
| (2) | () | $6 - 2\epsilon + f(N)\epsilon + \mathcal{O}(\epsilon^2)$ |

TABLE III. Examples of states with four scalar fields and two derivatives that have energies constrained to be equal at $N = 1$ in the critical $O(N)$ model in $4 - \epsilon$ space-time dimensions. The function $f(N) = (44 + 9N - \sqrt{624 - 8N + 9N^2})/(6(N + 8))$ and has $f(N = 1) = 14/27$.

We highlight the fact that the scalar irrep $\lambda = ()$ is a valid irrep for any integer $N$, i.e. it never belongs to one of the cases $\lambda^0$ or $\lambda^\pm$. Therefore, whenever we find an evanescent scalar state, the possibility of direct evanescence is excluded, and there is necessarily a constraint on the spectrum.

Note that it is not necessary for all of the $\lambda^+$ contributions to be annihilated by $\lambda^-$ contributions. Some could remain, and in this case, the state in the original irrep $\lambda^+$ has not gone null: it contributes to the partition function as the irrep $\lambda$ that it specializes as at that integer value of $N$. A simple example of this kind is the operator $\phi_1^{[i}\phi_2^{j]}$ formed by two distinct vectors $\phi_1^i, \phi_2^i$, which transforms in the $(1,1)$ irrep for integer $N \geq 4$, but as a vector in the special case of $N = 3$, via the cross product. The point is that the essence of the state is a $(1,1)$ irrep, despite its isolated specialization as a vector at $N = 3$.

The above discussions of constraints on the spectrum are very general, without any assumptions on how the partition function may be constructed as tensor products of fundamental objects that transform in some irreps of $O(N)$. If we do consider this scenario, however, we can gain a better understanding about precisely how the constraints arise.

For example, consider the tensor product of four distinct vector irreps $\phi_1^i\phi_2^j\phi_3^k\phi_4^l$, which has the following irrep decomposition in the large $N$ limit

$$\square \otimes \square \otimes \square \otimes \square = \left( \square\square\square\square \oplus \square\square \oplus \cdot \right) \oplus 3\left( \begin{smallmatrix}\square\square\square\\\square\end{smallmatrix} \oplus \begin{smallmatrix}\square\\\square\end{smallmatrix} \oplus \square\square \right)$$
$$\oplus 2\left( \square\square \oplus \square\square \oplus \cdot \right) \oplus 3\left( \begin{smallmatrix}\square\square\\\square\end{smallmatrix} \oplus \begin{smallmatrix}\square\\\square\end{smallmatrix} \right) \oplus \begin{smallmatrix}\square\\\square\\\square\end{smallmatrix}. \qquad (19)$$

The grouping is in light of the $SU(N)$ tensor product decomposition algebra in Eq. (5c). For example, the direct sum $(2,2) \oplus (2) \oplus ()$ can be viewed as stemming from restricting the $SU(N)$ irrep $(2,2)$ to $O(N)$. This is useful because, although there is no $SU(N)$ symmetry, the constraints occur between states within each grouping, and can be understood from the explicit construction of the $SU(N)$ operator and its subsequent restriction. (At the

| Lorentz irrep | $O(N)$ irrep $\lambda$ | $\Delta(N)$ |
|:---:|:---:|:---:|
| () | (2, 2) | $6 - 2\epsilon + \frac{6}{N+8}\,\epsilon + \mathcal{O}(\epsilon^2)$ |
| () | (2) | $6 - 2\epsilon + \frac{N+4}{N+8}\,\epsilon + \mathcal{O}(\epsilon^2)$ |
| (1, 1) | (2, 1, 1) | $6 - 2\epsilon + \frac{4}{N+8}\,\epsilon + \mathcal{O}(\epsilon^2)$ |
| (1, 1) | (1, 1) | $6 - 2\epsilon + \frac{N+2}{N+8}\,\epsilon + \mathcal{O}(\epsilon^2)$ |
| (1, 1) | (2, 2) | $6 - 2\epsilon + \frac{14}{3(N+8)}\,\epsilon + \mathcal{O}(\epsilon^2)$ |
| (1, 1) | (2) | $6 - 2\epsilon + g(N)\,\epsilon + \mathcal{O}(\epsilon^2)$ |

TABLE IV. Examples of states with four scalar fields and two derivatives that are have energies constrained to be equal at $N = 2$ in the critical $O(N)$ model in $4 - \epsilon$ space-time dimensions. The function $g(N)$ is given as the root of a cubic equation, with $g(N = 2) = 7/15$.

level of characters, the restriction is understood through the "folding" of a Dynkin diagram [26].)

Finally, we give some explicit examples of the constraints in the critical $O(N)$ model of scalar fields. Working perturbatively at the fixed point in $4 - \epsilon$ space-time dimensions, we consider the energies of the states corresponding to operators with four $\phi$ fields and two derivatives. Some examples are shown in Tables III and IV, with data taken from [19], where we group $O(N)$ irreps whose contributions to the partition function annihilate at $N = 1$ and $N = 2$, respectively. The annihilation between the $(2, 2)$ and $()$ irreps at $N = 1$, and between the $(2, 2)$ and $(2)$ irreps at $N = 2$ are examples of the constraint type $\Delta_{\lambda^-,i}(N) = \Delta_{\lambda,j}(N)$ in Eq. (18). The annihilation between the $(2, 1, 1)$ and $(1, 1)$ irreps at $N = 2$ is an example of the constraint type $\Delta_{\lambda^-,i}(N) = \Delta_{\lambda^+,k}(N)$ in Eq. (18), because at $N = 2$ their continued characters specialize as $\mp\chi_{()}^{O(2)}$ respectively; see Table II. For an example of direct evanescence, note that the state $(2, 2)$ will simply drop out of the spectrum at $N = 3$. As an example of a leftover $\lambda^+$ state, note that at $N = 5$ the $(2, 1, 1)$ has an isolated specialization as a $(2, 1)$ irrep of $O(5)$, with energy $6 - 2\epsilon + \frac{4}{13}\epsilon$.

## IV. OUTLOOK

It will be interesting to systematically map out the full set of constraints in the spectra of the $O(N)$ model, as well as to observe the constraints beyond leading order in the $\epsilon$ expansion, and non-perturbatively. An immediate question is the extent to which this can constrain the spectrum. Can they provide useful input to a bootstrap of the theory? Related to this question is the fact that we never really utilized the $O(N)$ symmetry at non-integer values, which fixes the functional form of the energies at non-integer $N$.

We have only discussed the integer spin irreps of $O(N)$

in this paper. The continuation of the spinor irreps and the consequent constraints on fermionic theories will be presented elsewhere.

The representation theory of $Sp(N)$ can be continued [11, 12], and the specialization rules of the characters are known [22], suggesting that similar constraints between states of theories with continued global $Sp(N)$ symmetries can be studied.

Another direction we leave for future work is to explore whether similar conditions could arise for the coefficients of analytically continued operator product expansion equations. This would be satisfying as it would make use of the continuation of the full representation theory, and not just the character theory.

Finally, the nature of the constraints being between different $O(N)$ irreps is curiously reminiscent of the degeneracies between different $SO(3)$ irreps in the Hydrogen atom. These degeneracies were ultimately explained by an $SO(4)$ symmetry. Could there be a (possibly generalized) symmetry understanding of the present evanescent-degeneracies, too?

## ACKNOWLEDGMENTS

We thank Sridip Pal for many illuminating discussions throughout the course of this work, and we thank Sridip Pal and Yuji Tachikawa for comments on a draft version of the manuscript. W.C. is supported by the Global Science Graduate Course (GSGC) program of the University of Tokyo and the JSPS KAKENHI grant numbers JP19H05810, JP20H01896, 20H05860, JP22J21553 and JP22KJ1072. W.C. also acknowledges the USTEP exchange program of the University of Tokyo and the FUTI scholarships for mid-to long-term studies from Friends of UTokyo, Inc. X.L. is supported by the U.S. Department of Energy under grant number DE-SC0009919. T.M. is supported by the World Premier International Research Center Initiative (WPI) MEXT, Japan, and by JSPS KAKENHI grants JP19H05810, JP20H01896, JP20H00153, and JP22K18712.

## Appendix A: Formula for the continued $O(N)$ character

In this appendix, we reproduce the explicit expressions from [22] for the function $F_\lambda$ appearing in Eq. (10). For convenience, we introduce the shorthand notation $\chi_V(x)$ for the ordinary character of the vector irrep

$$\chi_V(x) \equiv \chi_{(1)}^{O(N)}(x)\,, \qquad (A1)$$

and $p_n(x)$ for that of its $n$-th symmetric product:

$$p_n(x) \equiv \chi_{\text{sym}^n(1)}^{O(N)}(x)\,. \qquad (A2)$$

It is understood that both $\chi_V(x)$ and $p_n(x)$ depend on $N$, implicitly. One can compute $p_n(x)$ from $\chi_V(x)$ via

the bosonic plethystic exponential

$$\sum_{n=0}^{\infty} u^n \, p_n(x) = \exp\left[\sum_{k=1}^{\infty} \frac{1}{k} \, u^k \, \chi_V(x^k)\right], \qquad \text{(A3)}$$

which leads to the expression

$$p_n(x) = \frac{1}{n!} \, B_n(y_1, \cdots, y_n), \quad y_k \equiv \Gamma(k) \, \chi_V(x^k), \quad \text{(A4)}$$

with $B_n$ denoting the complete exponential Bell polynomials. From this it is clear that $p_n(x)$ is a function of $\chi_V(x^q)$ up to $q = n$:

$$p_n(x) = p_n\left[\chi_V(x), \, \chi_V(x^2), \, \cdots, \, \chi_V(x^n)\right]. \quad \text{(A5)}$$

Some explicit expressions are

$$p_0(x) = 1, \qquad \text{(A6a)}$$

$$p_1(x) = \chi_V(x), \qquad \text{(A6b)}$$

$$p_2(x) = \frac{1}{2}\left[\chi_V^2(x) + \chi_V(x^2)\right], \qquad \text{(A6c)}$$

$$p_3(x) = \frac{1}{6}\left[\chi_V^3(x) + 3\chi_V(x)\,\chi_V(x^2) + 2\chi_V(x^3)\right], \qquad \text{(A6d)}$$

$$p_4(x) = \frac{1}{24}\left[\chi_V^4(x) + 6\chi_V^2(x)\,\chi_V(x^2) + 8\chi_V(x)\,\chi_V(x^3)\right.$$
$$\left. + 3\chi_V^2(x^2) + 6\chi_V(x^4)\right]. \qquad \text{(A6e)}$$

For convenience, let us also define $p_n(x) = 0, \ \forall \, n < 0$.

The continued $O(N)$ character can be written as

$$\overline{\chi}_\lambda^{O(N)}(x) = \det\left[p_{\lambda_i - i + j}(x) - p_{\lambda_i - i - j}(x)\right]. \qquad \text{(A7)}$$

Combining this with Eq. (A4), one can obtain an explicit expression for any $\overline{\chi}_\lambda^{O(N)}(x)$ in the form of Eq. (10). For example, $F_{(1,1)}$ and $F_{(2,2)}$ have the following expressions

$$\overline{\chi}_{(1,1)}^{O(N)}(x) = \det\begin{bmatrix} p_1(x) - p_{-1}(x) & p_2(x) - p_{-2}(x) \\ p_0(x) - p_{-2}(x) & p_1(x) - p_{-3}(x) \end{bmatrix}$$

$$= \tfrac{1}{2}\,\chi_V^2(x) - \tfrac{1}{2}\,\chi_V(x^2), \qquad \text{(A8a)}$$

$$\overline{\chi}_{(2,2)}^{O(N)}(x) = \det\begin{bmatrix} p_2(x) - p_0(x) & p_3(x) - p_{-1}(x) \\ p_1(x) - p_{-1}(x) & p_2(x) - p_{-2}(x) \end{bmatrix}$$

$$= \tfrac{1}{12}\,\chi_V^4(x) + \tfrac{1}{4}\,\chi_V^2(x^2) - \tfrac{1}{3}\,\chi_V(x)\,\chi_V(x^3)$$
$$- \tfrac{1}{2}\,\chi_V^2(x) - \tfrac{1}{2}\,\chi_V(x^2). \qquad \text{(A8b)}$$

## Appendix B: Specialization rules of the continued characters by clipping Young diagrams

In this appendix, we provide prescriptions of clipping Young diagrams for determining how a continued character $\overline{\chi}_\lambda^{SU(N)}$ or $\overline{\chi}_\lambda^{O(N)}$ specializes to an ordinary one. This is an alternative to the Young diagram based method appearing in [22].

### 1. $SU(N)$ characters — West Coast clipping

The specialization rule for $\overline{\chi}_\lambda^{SU(N)}$ is quite straightforward and is given by Eq. (6). Despite its simplicity, it is useful to visualize this specialization rule using Young diagrams. Specifically, starting from a given Young diagram $\lambda$ and an integer $N$, one can figure out the right-hand side of Eq. (6) through an algorithmic prescription of *clipping off the West Coast* of $\lambda$, which has the following steps:

1. **Identify the West Coast:** We call the leftmost column of a Young diagram $\lambda$ its *West Coast*. As some explicit examples, the colored boxes in Eq. (B1) form the West Coast of $\lambda = (3, 3, 2)$, and the colored boxes in Eq. (B3) form the West Coast of $\lambda = (2, 2, 1)$. If $\lambda$ has an empty West Coast, it must be an empty Young diagram with zero number of boxes, $\lambda = ()$. In this case, $\lambda$ denotes the trivial irrep of $SU(N)$, a valid irrep for any positive integer $N$; we stop and return $\overline{\chi}_\lambda^{SU(N)} = \chi_\lambda^{SU(N)}$. Otherwise, we continue to the next step.

2. **Determine whether to clip:** Count the number of boxes $n_W$ on the West Coast of $\lambda$. This equals the length of the partition $n_W = l(\lambda)$. If $n_W < N$, $\lambda$ is a valid irrep for $SU(N)$ and we do not need to clip; we stop and return $\overline{\chi}_\lambda^{SU(N)} = \chi_\lambda^{SU(N)}$. If $n_W > N$, we also do not need to clip; we stop and return $\overline{\chi}_\lambda^{SU(N)} = 0$. If $n_W = N$, we continue to the next step.

3. **Clip and repeat the steps:** We clip off (remove) the West Coast of the Young diagram $\lambda$ to obtain a new Young diagram $\lambda_{\text{new}}$, and return $\overline{\chi}_\lambda^{SU(N)} = \overline{\chi}_{\lambda_{\text{new}}}^{SU(N)}$. We then repeat the above steps with $\lambda_{\text{new}}$ as the new specified partition, and recurse until the algorithm stops.

With the above algorithm, we will eventually end up with either $\overline{\chi}_\lambda^{SU(N)} = 0$, or $\overline{\chi}_\lambda^{SU(N)} = \chi_{\lambda'}^{SU(N)}$ for some valid irrep $\lambda'$ of $SU(N)$. Clearly, the ultimate output will agree with Eq. (6).

It is useful to visualize the above West Coast clipping prescription by an explicit example. For this purpose, let us consider $\overline{\chi}_{(3,3,2)}^{SU(3)}$, where $\lambda = (3, 3, 2)$ and $N = 3$. Following the steps in the prescription, we find the first

round of the clipping to be

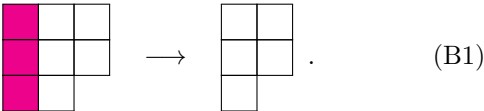

$$\longrightarrow \qquad . \qquad (B1)$$

It gives us a new Young diagram $(2, 2, 1)$, which means

$$\overline{\chi}^{SU(3)}_{(3,3,2)} = \overline{\chi}^{SU(3)}_{(2,2,1)} . \qquad (B2)$$

Following the last step in the prescription, we now use this new Young diagram $(2, 2, 1)$ as input to repeat the clipping steps. We find the second round of clipping

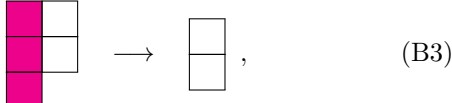

$$\longrightarrow \qquad , \qquad (B3)$$

leading us to another new Young diagram $(1, 1)$ and

$$\overline{\chi}^{SU(3)}_{(2,2,1)} = \overline{\chi}^{SU(3)}_{(1,1)} . \qquad (B4)$$

Now using $(1, 1)$ as new input again to repeat the clipping steps, we finally find that there is no need to clip further, because $n_W = 2 < N = 3$; the algorithm returns

$$\overline{\chi}^{SU(3)}_{(1,1)} = \chi^{SU(3)}_{(1,1)} . \qquad (B5)$$

Putting Eqs. (B2), (B4) and (B5) together, we get

$$\overline{\chi}^{SU(3)}_{(3,3,2)} = \overline{\chi}^{SU(3)}_{(2,2,1)} = \overline{\chi}^{SU(3)}_{(1,1)} = \chi^{SU(3)}_{(1,1)} , \qquad (B6)$$

which agrees with Eq. (6).

### 2. $O(N)$ characters — East Coast clipping

The specialization rule for $\overline{\chi}^{O(N)}_\lambda$ is not as simple as that for $\overline{\chi}^{SU(N)}_\lambda$ in Eq. (6). One way to figure it out is to employ the formula in Eq. (10), but this soon becomes cumbersome for large partitions. Here, we present an algorithmic prescription of *clipping off the East Coast* of $\lambda$, which has the following steps:

1. **Identify the East Coast:** We call the rightmost shell of a Young diagram $\lambda$ its *East Coast*. As some explicit examples, the gray boxes in Eq. (B9) form the East Coast of $\lambda = (5, 4, 3, 3, 3, 2, 1)$, and the gray boxes in Eq. (B11) form the East Coast of $\lambda = (5, 4, 3, 3, 1, 1, 1)$. If $\lambda$ has an empty East Coast, it must be an empty Young diagram with zero number of boxes, $\lambda = ()$. In this case, $\lambda$ denotes the trivial irrep of $O(N)$, a valid irrep for any positive integer $N$; we stop and return $\overline{\chi}^{O(N)}_\lambda = \chi^{O(N)}_\lambda$. Otherwise, we continue to the next step.

2. **Identify the clipping center:** We label each box on the East Coast with an integer $s = i - j$, where $i$ and $j$ are the row and column coordinates of the box. Explicit examples are shown in Eqs. (B9) and (B11). This integer spans the range $1 - \lambda_1 \leq s \leq l(\lambda) - 1$, and increases by one in each step of moving south or west along the East Coast. We identify the box with $s = r$, and call it the "clipping center", which is the box colored in darker gray in Eqs. (B9) and (B11). If the center cannot be found, then we must have $r > l(\lambda) - 1$, meaning that $\lambda$ is a valid irrep of $O(N)$; we stop and return $\overline{\chi}^{O(N)}_\lambda = \chi^{O(N)}_\lambda$. Otherwise, we continue to the next step.

3. **Identify the clipping patch:** All the boxes strictly below the clipping center are included in the clipping patch. In addition, we also include the clipping center, as well as another $n_A$ boxes along the East Coast that are towards the east or north from the center. The number $n_A$ is determined as

$$n_A = n_B + \frac{1 + (-1)^N}{2} , \qquad (B7)$$

where $n_B$ denotes the number of boxes strictly below the clipping center. In the examples of Eqs. (B9) and (B11), we have $n_A = n_B = 1$ and $n_A = n_B = 3$ respectively, and the resulting clipping patches are colored in magenta.

4. **Clip and repeat the steps:** We clip off (remove) the clipping patch from the Young diagram $\lambda$. If the resulting shape is not a Young diagram, we return $\overline{\chi}^{O(N)}_\lambda = 0$. Otherwise, we get a new Young diagram $\lambda_{\text{new}}$ from the clipping, and return

$$\overline{\chi}^{O(N)}_\lambda = (-1)^{n_{\text{rows}}+N} \overline{\chi}^{O(N)}_{\lambda_{\text{new}}} , \qquad (B8)$$

where $n_{\text{rows}}$ denotes the number of rows that the clipping patch spans. In Eqs. (B9) and (B11), $n_{\text{rows}} = 2$ and $n_{\text{rows}} = 5$, respectively. We then repeat the above steps with $\lambda_{\text{new}}$ as the new specified partition, and recurse until the algorithm stops.

With the above algorithm, we will eventually end up with either $\overline{\chi}^{O(N)}_\lambda = 0$, or $\overline{\chi}^{O(N)}_\lambda = \pm\chi^{O(N)}_{\lambda'}$ for some valid irrep $\lambda'$ of $O(N)$.

To visualize the above East Coast clipping prescription, let us consider the example $\overline{\chi}^{O(7)}_{(5,4,3,3,3,2,1)}$, where $\lambda = (5, 4, 3, 3, 3, 2, 1)$ and $N = 7$; the rank is $r = 3$. Following the steps in the prescription, we find the first round of the clipping

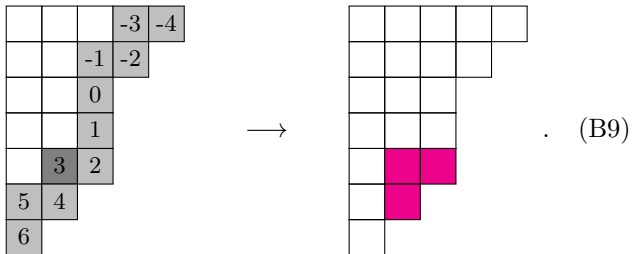

$$\longrightarrow \qquad . \qquad (B9)$$

It gives us a new Young diagram $(5, 4, 3, 3, 1, 1, 1)$. This clipping has $n_{\text{rows}} = 2$, which according to Eq. (B8), gives us a negative overall sign:

$$\overline{\chi}^{O(7)}_{(5,4,3,3,3,2,1)} = -\overline{\chi}^{O(7)}_{(5,4,3,3,1,1,1)}.$$ (B10)

Following the last step in the prescription, we now use this new Young diagram $(5, 4, 3, 3, 1, 1, 1)$ as input to repeat the clipping steps. We find the second round of clipping

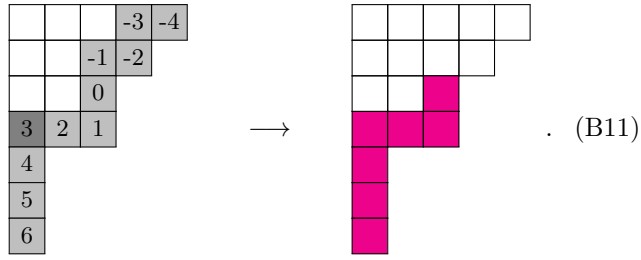

. (B11)

It gives us a new Young diagram $(5, 4, 2)$. This clipping

has $n_{\text{rows}} = 5$, which according to Eq. (B8), gives us a positive overall sign:

$$\overline{\chi}^{O(7)}_{(5,4,3,3,1,1,1)} = \overline{\chi}^{O(7)}_{(5,4,2)}.$$ (B12)

Now using $(5, 4, 2)$ as new input again to repeat the clipping steps, we finally find that there is no need to clip further, because no clipping center can be found; the algorithm returns

$$\overline{\chi}^{O(7)}_{(5,4,2)} = \chi^{O(7)}_{(5,4,2)}.$$ (B13)

Putting Eqs. (B10), (B12) and (B13) together, we get the final output of the prescription

$$\overline{\chi}^{O(7)}_{(5,4,3,3,3,2,1)} = -\chi^{O(7)}_{(5,4,2)}.$$ (B14)

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
