# Peer review of "Constraints on the spectrum of field theories with non-integer $O(N)$ symmetry from quantum evanescence"

_SciPost Physics Core_

## Round 1 · Referee Report · Anonymous (Referee 2) · 2024-4-22

Report

I find the revisions, along with the authors' reply to the three referee reports, very helpful in better understanding the paper. The sharpened terminology and added comments serve to clarify the underlying philosophy, the role played by the non-integrality of N, and the aspects of the paper that are novel. Under a strict view that considers only unitary, integer-N theories, the relationships between states that form the subject of this paper are imperceptible, but continuation of integer-valued physical parameters is a commonplace procedure, which it is worth understanding at the deepest possible level.

Recommendation

Publish (meets expectations and criteria for this Journal)

  • validity: -
  • significance: -
  • originality: -
  • clarity: -
  • formatting: -
  • grammar: -

Author:  Tom Melia  on 2024-06-25  [id 4584]

(in reply to Report 1 on 2024-04-22)

We thank the referee for their reading of the resubmission and their comments

---

## Round 1 · Referee Report · Anonymous (Referee 3) · 2024-5-10

Report

Generally I think the paper is mostly acceptable. However while
the word degeneracy is no longer in the title it appears frequently in
the introduction. I thought it was agreed that what washing described was not degeneracy in the usual sense but a necessary consistency condition. In my view the introductory paragraphs should be modified to reflect this

Secondly the results in (14a,b,c) and in Table II are a direct reflection
of the Racah Speiser algorithm. For what it is worth a concise discussion of this is given in an appendix to hep-th/0209056. Similar results can be found in maths textbooks.

With some revision this paper would be acceptable.

Recommendation

Ask for minor revision

  • validity: -
  • significance: -
  • originality: -
  • clarity: -
  • formatting: -
  • grammar: -

Author:  Tom Melia  on 2024-06-25  [id 4583]

(in reply to Report 2 on 2024-05-10)
Category:
reply to objection

We thank the referee for re-reading the submission and for highlighting some remaining sources of confusion regarding the Racah Speiser algorithm. Despite some similarities with 0s and +/- signs appearing in the algorithm, there is no direct nor indirect connection to our specialization rules or the results in (14a,b,c) or Table II. The RS algorithm implements a completely different mathematical task. We have provided a detailed footnote in the latest version to clarify this.

We would also like to reiterate that the introductory paragraphs were in fact modified to reflect the fact that what is being described is a new phenomena - not degeneracy in the usual sense. This is the content of paragraph 6.

We hope this clears up any remaining miscommunications.

---

## Round 1 · Referee Report · Anonymous (Referee 1) · 2024-5-20

Report

I think the changes implemented by the authors improve the clarity of the paper and I am happy with the paper being published as is

Recommendation

Publish (meets expectations and criteria for this Journal)

  • validity: -
  • significance: -
  • originality: -
  • clarity: -
  • formatting: -
  • grammar: -

Author:  Tom Melia  on 2024-06-25  [id 4585]

(in reply to Report 3 on 2024-05-20)

We thank the referee for their reading of the resubmission and their comment

---

## Round 1 · Author Response

We have made major revisions to the text in response to the referee reports with a list of changes as detailed below.

---

## Round 1 · List of Changes

In light of the comments of referees and editor, we have made the following major revisions to the manuscript.

  1. To address the issue of the usage of the term degeneracy we have opted to instead describe the phenomena as a "constraint" on energy spectra, replacing the usage of 'degeneracy' everywhere it appeared in the original text. We have changed the title to "Constraints on the spectrum of field theories with non-integer O(N) symmetry from quantum evanescence". We have also added/made revisions to what are now the 4th, 5th, and 6th paragraphs, and to the abstract, described in the following:

a) In light of the comment of referee 3, we explicitly emphasize that certain representations are "becoming negative" at integer N, and that we are identifying thus the fact that some (not all) evanescent operators have to drop out pairwise. This is a result we are not aware of in existing literature. It is of course, as we outlined in our response to referee 3, the key ingredient for obtaining the spectrum constraints. We emphasize, in light of what we believe may be a miscommunication between us and referee 2 and 3, that not all evanescent operators drop out pairwise and thus lead to constraints. Phrasing in terms of the viewpoint of referee 2, that operators are becoming linearly related at certain N, we point out such linear relations do not in general lead to the existence of the constraints we identify.

b) We added a paragraph to highlight the difference of this constraint phenomena from usual degeneracy. Namely, that states become degenerate and simultaneously drop out of the spectrum at precisely integer N, coining this behaviour as 'evanescent-degeneracy'. 

We hope that this addresses the referee and editors concerns, making clear the difference to the usual usage of degeneracy.

  1. To address another of the points of referee 2, we added a clarifying sentence above eq 15 that states the continued partition function is valid for all real values of N, same as the continued characters and the continued tensor product decomposition algebra.

  2. We addressed the misuse of coupling constant to describe N as pointed out by referee 1 in the 3rd paragraph.

We hope that these revisions can satisfy the referees' concerns, clear up some potential miscommunications, and result in the publication of the manuscript.

---

## Editorial Decision

resubmitted